# Organisational Impact of a Remote Patient Monitoring System for Heart Failure Management: The Experience of 29 Cardiology Departments in France

**DOI:** 10.3390/ijerph20054366

**Published:** 2023-02-28

**Authors:** Sarah Alami, Laurène Courouve, Guila Lancman, Pierrette Gomis, Gisele Al-Hamoud, Corinne Laurelli, Hélène Pasche, Gilles Chatellier, Grégoire Mercier, François Roubille, Cécile Delval, Isabelle Durand-Zaleski

**Affiliations:** 1ALSI, Air Liquide Santé International, 92220 Bagneux, France; 2CEMKA, 92340 Bourg-la-Reine, France; 3Department of Statistics Informatics and Public Health, Université Paris-Cité, 75006 Paris, France; 4Clinical Research Unit, Groupe Hospitalier Paris Saint Joseph, 75014 Paris, France; 5Economic Evaluation Unit (URME), University Hospital of Montpellier, 34295 Montpellier, France; 6IDESP, Université de Montpellier, INSERM, 34000 Montpellier, France; 7Cardiology Department, Hôpital Lapeyronie, PhyMedExp, University of Montpellier, INSERM, CNRS, CHRU, INI-CRT, 34090 Montpellier, France; 8Université de Paris, CRESS, INSERM, INRA, URCEco, AP-HP, Hôpital de l’Hôtel Dieu, 75004 Paris, France; 9Santé Publique Hôpital Henri Mondor, 51 Avenue du Maréchal de Lattre de Tassigny F, 94010 Créteil, France

**Keywords:** heart failure, remote patient monitoring, organisational impact, health technology assessment, ambulatory, emergency department

## Abstract

Remote patient monitoring (RPM) for the management of patients with chronic heart failure (CHF) has been widely studied from clinical and health-economic points of view. In contrast, data on the organisational impact of this type of RPM are scarce. The objective of the present study of cardiology departments (CDs) in France was to describe the organisational impact of the Chronic Care Connect^TM^ (CCC^TM^) RPM system for CHF. An organisational impact map for health technology assessment was used to identify and define the criteria evaluated in the present survey, including the care process, equipment, infrastructure, training, skill transfers, and the stakeholders’ abilities to implement the care process. In April 2021, an online questionnaire was sent to 31 French CDs that were using CCC^TM^ for CHF management: 29 (94%) completed the questionnaire. The survey results showed that CDs progressively modified their organisational structures upon or shortly after the implementation of the RPM device. Twenty-four departments (83%) had created a dedicated team, sixteen (55%) had provided dedicated outpatient consultations for patients with an emergency alert, and twenty-five (86%) admitted patients directly (i.e., avoiding the need to attend the emergency department). The present survey is the first to have assessed the organisational impact of the implementation of the CCC^TM^ RPM device for CHF management. The results highlighted the variety of organisational structures, which tended to structure with the use of the device.

## 1. Introduction

Chronic heart failure (CHF) is a global pandemic that currently affects the lives of 64 million people worldwide [1] and 2.3% of the French population [2]. The condition is characterised by a high mortality rate in the post-discharge period and a high likelihood of hospital readmission for acute, decompensated heart failure [3,4]. Remote patient monitoring (RPM) systems (such as Chronic Care Connect^TM^ (CCC^TM^) e-health solution) are now emerging as additional tools for CHF care management.

In France, a national scheme for promoting and funding RPM (Expérimentations de Télémédecine pour l’Amélioration des Parcours en Santé (ETAPES)) was launched in 2014. For inclusion in ETAPES, RPM systems must meet strict specifications and must combine a digital device, medical telemonitoring, and therapeutic support [5]. Although an evaluation of the device’s clinical benefit, impact on quality of life, and economic impact is obligatory, other consequences must also be explored—notably with regard to access to care, quality of care, and care organisation [6]. Until now, these evaluations were mainly focused on assessing the medical and economic impacts [7]. The assessment of a device’s organisational impact has lagged behind the other assessments because of the lack of a specific methodology and guidelines [8]—even though this is essential for fully appraising the influence of medical technologies [9].

In order to tackle this problem and include the organisational impact in the assessment process, the French National Authority for Health (Haute Autorité de Santé (HAS)) published a guide in December 2020 [10]. The guide gave a definition of organisational impact (“an effect or consequence of the health technology on the characteristics and functioning of an organisation involved in the care process or the user’s life pathway”) and suggested criteria for measuring and justifying the RPM device’s effects in this respect. The organisational impact has now been included in the criteria evaluated by the French National Medical Device and Health Technology Evaluation Committee (Commission Nationale d’Evaluation des Dispositifs Médicaux et Technologies de Santé (CNEDiMTS)) [11].

The organisational impact can be documented from various perspectives (that of the patient, the healthcare professional, the healthcare system, the hospitals, etc.) and by using various methods and data sources. The objective of the present study was to describe the organisational impact of CCC^TM^ on CHF management from the perspective of healthcare professionals using the device.

## 2. Methods

### 2.1. Survey Design

An online survey was conducted among 31 French cardiology departments (CDs) known to use CCC^TM^ for CHF management. The questionnaire was sent to all 31 public- or private-sector institutions known to have monitored at least 20 patients with CCC^TM^ between 2018 and 2020.

Data were collected between 16 April and 26 April 2021 using an online questionnaire. In each CD, only one questionnaire was filled; the survey was designed to be completed by the team as a whole.

### 2.2. Chronic Care Connect^TM^

Chronic Care Connect^TM^ is a remote patient monitoring (RPM) solution for heart failure (HF) management composed of a class IIA medical device and non-medical human assistance. A connected scale allows daily weight collection, and a mobile application allows patients recording of HF symptoms. One of the CCC^TM^’s unique features is the integration of this new, non-hospital-based stakeholder: a monitoring centre whose nurses are specially trained in remote monitoring and initially screen the alerts received. If an alert is judged to be relevant, it is sent to the patient’s CD.

### 2.3. Study Measurements and Outcomes

The entire questionnaire was based on the organisational impact map for health technology assessment when the standards were applicable [10]. This mapping process is a structured way of identifying and quantifying organisational impacts, in which three macro-criteria are divided into several sub-criteria.

For macro-criterion 1, the various sub-criteria covered time consumption, the speed or duration of care processes, the equipment and infrastructures used in the RPM process, and the skills needed so that the stakeholders could implement the care process. For macro-criterion 2, the sub-criteria were related to stakeholder training and skill transfers. Lastly, for macro-criterion 3, the sub-criteria covered impacts related to communication, society, and the environment.

For each sub-criterion, an assessment of the potential impact of CCC^TM^ was performed (Table 1). When the sub-criterion was not relevant for CCC^TM^, “not applicable” was stated in the questionnaire. To assess the organisational impact of each sub-criterion, several data sources and methods are used, including this survey among health professionals. All the results only concern the organisational impact of CCC^TM^.

### 2.4. Statistical Analysis

Most of the statistical analyses were descriptive. All statistical analyses were performed with SAS^®^ software (version 9.4, SAS^®^ Institute Inc., Cary, NC, USA).

## 3. Results

### 3.1. Participants

A link to the online survey was sent to the 31 CDs equipped with the CCC^TM^ device and which had sufficient experience of its use (i.e., RPM with at least 20 patients). Of the 31 CDs contacted, 29 (94%) completed the study questionnaire. In most CDs, several different healthcare professionals answered the survey: most were cardiologists (27 out of 29) and nurses (18 out of 29). The response rates per question were 100% unless otherwise stated.

### 3.2. Characteristics of the Participating Cardiology Departments

From the system’s date of deployment in France (1 February 2018) to the survey closure date (26 April 2021), the 29 participating CDs monitored 63% of the patients remotely monitored by CCC^TM^ over the same time frame. On average, 122 patients per CD were monitored (Table 2). Twenty-four of the CDs (83%) were in public-sector institutions, and five (17%) were in private-sector institutions. The CDs had used the device for an average of 23 months; 14 (48%) had used it for more than 24 months, and 7 (24%) had used it for less than 12 months.

### 3.3. Impacts of the Health Technology on the Care Process (Macro-Criterion 1)

#### 3.3.1. Impact on the Initiation of the Care Process

Healthcare professionals were asked about the time interval between cardiac decompensation and the initiation of medical care. Most of the respondents declared that RPM for patients with CHF was associated with a shorter time interval: 21 of the 29 (72%) CDs “absolutely agreed” and 7 (24%) “mostly agreed”.

#### 3.3.2. Impact on the Pace or Duration of the Process

The use of RPM might modify the process pace and overall duration of the care process as it includes alert management, which necessitates additional time. Alert management includes several phases, acknowledging the alert, making a diagnosis, responding to the alert, and triggering the intervention (Table 3). The survey results showed that nurses performed most of the tasks in alert management: nurses were primarily involved in overseeing the first phase (acknowledgement of the alert) in 23 of the 24 (96%) CDs, whereas cardiologists were primarily involved to a lesser extent (in 16 of the 29 CDs (55%)).

Twenty-two of the twenty-nine CDs (76%) chose to deal with the alerts Monday to Friday, during office hours. In the seven other CDs, the various management procedures were explained by the low number of remotely monitored patients. Depending on the number of patients and the type of healthcare professional involved, alert management may require dedicated time. In CDs with less than 50 patients being remotely monitored, the average time spent on this task was 4.1 h per week for nurses and 1.3 h per week for cardiologists. In CDs with more than 50 patients being remotely monitored, the average time spent was 14.3 h per week for nurses and 1.3 h per week for cardiologists.

Regarding the overall duration of the care process, RPM was initially prescribed for six months. In 28 of the 29 CDs (97%), this prescription was renewed. In 12 CDs (41%), the proportion of patients with a renewed prescription was over 80%. Renewal of the prescription was prompted by unstable disease (in 23 of the 29 CDs (79%)) or a patient request (again in 23 of the 29 (79%)).

#### 3.3.3. Impact on Process Timing or Content

The survey results demonstrated that all participating CDs had specifically changed their organisational structure in response to alert management and thus changed the care process content (Figure 1). Sixteen of the twenty-nine CDs (55%) had specifically allotted time for outpatient consultations after an emergency alert. Fourteen out of sixteen (88%) had started the consultations at the same time as (or shortly after) the introduction of RPM. Eighteen of the CDs (62%) had created an organisational structure dedicated to CHF medication titration [12]; only a third of the CDs had implemented it before RPM deployment. In 16 of the 18 CDs (88%), the medication was titrated during a face-to-face consultation. However, six of the 18 had the option to do it over the phone.

To avoid admission to the emergency department (ED), 25 of the 29 CDs (86%) set up a procedure for direct emergency admission to the CD. In 21 of the 25 CDs (84%), this system had been implemented at the same time as (or shortly after) the introduction of RPM.

#### 3.3.4. Impact on the Organisation of Human Resources

The implementation of RPM for patients with CHF requires qualified, trained human resources. In 24 of the 29 CDs (83%), a dedicated RPM team had been set up. Two CDs out of five (40%) stated that they were not monitoring enough patients for a dedicated team, and two others (40%) declared that they lacked funding.

Whether or not a team was assigned, in all questioned CDs, at least one cardiologist was involved in RPM, and at least one nurse was involved in 24 of the 29 CDs (83%). On average, 1.3 healthcare professionals (full-time equivalents (FTEs), and regardless of their role) per CD were involved in RPM (median: 1). For the period from June 2019 to June 2020, the mean number of remotely monitored patients per FTE healthcare professional was 74 (median: 38; range: 11–340).

#### 3.3.5. Impact on the Allocation of Materials and Equipment

Along with human resources, the survey also included questions on equipment. According to the healthcare professionals, the use of RPM did not require any additional equipment because CCC^TM^ relied on the CD’s existing computers (according to 18 of the 24 (75%) healthcare professionals who answered this item), and the CD had implemented a dedicated telephone line (according to 16 of the healthcare professionals). Only four CDs (14%) had funded the creation of a dedicated remote monitoring room.

#### 3.3.6. Impact on the Continuity of Care

The healthcare professionals were also asked about the impact of RPM on the continuity of care and mainly reported difficulties when team members were on leave. Twenty of the 29 (76%) respondents declared difficulties in ensuring care continuity: 16 (55%) encountered some difficulties, and 4 (14%) encountered many difficulties).

### 3.4. Impacts of the Health Technology on the Abilities and Skills Required of Stakeholders to Implement the Care Process (Macro-Criterion 2)

#### 3.4.1. Impact on the Skills Required of Stakeholders

To manage RPM in a CHF setting, for both healthcare professionals and patients training was required [13]. Patients were mainly trained to use the device (i.e., the smart scale and the tablet computer) but also had access to a disease management program supervised by a nurse to help them manage their disease daily.

According to the survey respondents, this training took an average of 4 h per patient over a 12-month period. Thus, training in disease management for 50 patients would take a total of 6 weeks per FTE healthcare professional.

Most of this training was delivered by nurses (in 22 of the 29 CDs (76%)). Disease management training was sometimes delivered by the device supplier (11 out of 29 (38%)), but this was mainly due to a lack of human resources in the CDs (mentioned by 10 out of 11 respondents (91%)).

In 19 of the 29 CDs (66%), some or all of the healthcare professionals involved in RPM received specific training. In some cases, this training was delivered as part of continuing professional development on CHF or disease management.

#### 3.4.2. Impact on Physician-to-Nurse Delegation of Duties

In 15 of the 29 CDs (52%), the organisational structure implemented for RPM allowed cardiologists to delegate certain medical procedures to nurses. In 6 CDs (21%), task delegation was part of a collaborative agreement with the health authorities: for example, nurses were allowed to manage alerts, refer patients, and perform follow-up consultations for medication titration in the absence of a cardiologist. In the other 9 CDs (31%), task delegation was mostly limited to alert screening (mentioned five times), the prescription of standard laboratory tests (mentioned three times) and changes in treatment supervised by a cardiologist (mentioned three times).

The device also demanded skills transfers: the nurses contributed significantly to the administrative work and the coding of medical procedures.

#### 3.4.3. Impact on the Coordination between the Stakeholders

##### A New, Non-Hospital-Based Stakeholder

One of CCC^TM^’s specific features is the introduction of a new, non-hospital-based stakeholder into the care process. Twenty-six of the twenty-nine (90%) survey participants stated that the existence of a monitoring centre with a dedicated team of nurses was the main reason for choosing CCC^TM^. The respondents considered that the alerts screened by the monitoring centre were relevant (mean relevancy score: 8.2 out of 10).

The equipment provided and the manufacturer’s experience in the field of RPM were also mentioned as factors that facilitated the implementation of this new organisational structure (according to 22 (76%) and 17 (59%) CDs, respectively).

##### Coordination between Ambulatory Care and Hospital Care

RPM requires coordination between hospital stakeholders and those in ambulatory care, including general practitioners (GPs). The respondents noted that GPs did not have direct access to the RPM software. The GPs were sent updates about the patients’ follow-up by the hospital staff—usually by e-mail (according to 22 of the 28 respondents (79%)) or by phone (22 out of 28 (79%)). The respondents stated that the implementation of RPM helped to improve coordination with primary care: 8 of the 29 (28%) agreed absolutely with this statement, and 11(38%) agreed somewhat.

##### Impact on Healthcare Professionals’ Working Conditions and the Patients’ Quality of Life

Fourteen of the twenty-nine (48%) respondents agreed that RPM could also impact the healthcare professionals’ working conditions. All fourteen considered that the organisational structure implemented for RPM helped to improve the healthcare professional-patient relationship, gave the healthcare professionals more independence, and increased their level of satisfaction. However, twelve of the fourteen (86%) respondents also reported a decrease in the speed and quantity of work to some extent.

According to the surveyed healthcare professionals, RPM had a positive impact on the patient’s quality of life and level of satisfaction: 16 of the 29 (55%) respondents considered that the quality of life was very much better, and 11 (38%) considered that quality of life was somewhat better.

## 4. Discussion

The present survey of hospital-based healthcare professionals’ opinions provided a description of the organisational implementation of CCC^TM^ for RPM for CHF management.

This survey stands out because it used a standardised method to directly address all the organisational impacts (whether positive or negative) perceived by a representative panel of professionals with significant expertise in CHF management and RPM. Furthermore, it should be possible to apply this method to other health technologies in other contexts. Whereas, in previous studies, the organisational impact was less explored and poorly documented, partially due to the lack of a standardised method. Most studies on the impact of health technologies have only referred to certain organisational impacts, such as the effects on human resources and organisational structures [14,15].

The results show that the CDs restructured their working practices over time. Most had implemented a new organisational structure upon or shortly after the implementation of RPM for decompensations of CHF. Twenty-four of the CDs (83%) had created a dedicated RPM team. More than half of the CDs had created specific outpatient consultations for patients with an emergency alert; in fourteen of the sixteen (88%), this initiative was taken upon or shortly after the introduction of RPM. Eighteen of the twenty-nine CDs (62%) had introduced specific procedures for medication titration; twelve of the eighteen (67%) did this after RPM had been implemented. Twenty-five of the twenty-nine CDs (86%) admitted patients directly and thus avoided an ED admission. Even though the CDs had changed their organisational structure over time, the survey results highlighted differences from one department to another. These differences were mainly related to each team’s level of familiarity with RPM and the number of remotely monitored patients. Healthcare professionals were generally satisfied with the introduction of a new stakeholder (a monitoring centre with a dedicated team of nurses) for some care management tasks. Moreover, RPM does not demand a substantial investment in equipment; only computers and a phone line are needed to manage the patients’ alerts. However, in the short term, RPM requires dedicated staff time to be set aside, and this can differ from one healthcare professional to another. Cardiologists spent an average of 2 h a week on alert management, whereas nurses spent an average of 10 h a week on the same task. Nevertheless, the literature data show that in the long term, RPM with an appropriate organisational structure is associated with shorter lengths of stay in the ED and in hospitals in general, which might free up time for patients and hospital staff [16]. Our present survey also demonstrated that RPM led to more training and more skill transfer for the participating healthcare professionals. Lastly, the healthcare professionals stated that the changes in organisational structure improved the quality of patient care. Most of the respondents reported that care provision took less time, with a smoother care pathway and better quality of life for the patients.

The main limitation of the present study is embedded in its design; the results reflected the opinions of hospital-based healthcare professionals of a relatively small sample of French CDs. Indeed, for this study, only CDs already using CCC^TM^ were targeted, which is already limiting the survey pool. Among these CDs using the device, only those with at least 20 patients were included. Although it was the most relevant choice in this situation, the results cannot be extrapolated to RPM devices in general. Moreover, as there is no scoring system in the questionnaire, it is impossible to have a hierarchy of the mentioned organisational impacts. Secondly, some of the HAS’ criteria could not be adapted for the assessment of CCC^TM^. The “mapping” technique is a robust tool for identifying organisational impacts but cannot be used directly to assess health technology devices. Lastly, impacts that were not thought to be relevant to CCC^TM^ were not assessed (i.e., macro-criterion 3).

Given the great variety of organisational structures highlighted by our survey, it will now be necessary to assess the corresponding clinical and economic impacts on patients with CHF and to disseminate the best-performing practices. To this end, a quantitative study is currently performed using the French national healthcare database (Système National des Données de Santé), which should complement the qualitative data on healthcare professionals’ perceptions collected in the present study. The quantitative study will enable the assessment of other organisational impacts (such as the hospital readmission rate and the lengths of stay in the ED or in the hospital, as the clinical and economic impacts) as a function of the CD’s organisational structure and human resources.

## 5. Conclusions

The present survey is the first to have assessed the organisational impact of the implementation of the CCC^TM^ RPM device for CHF management. The results highlighted the variety of organisational structures, which tends to structure with the use of the device.

## Figures and Tables

**Figure 1 ijerph-20-04366-f001:**
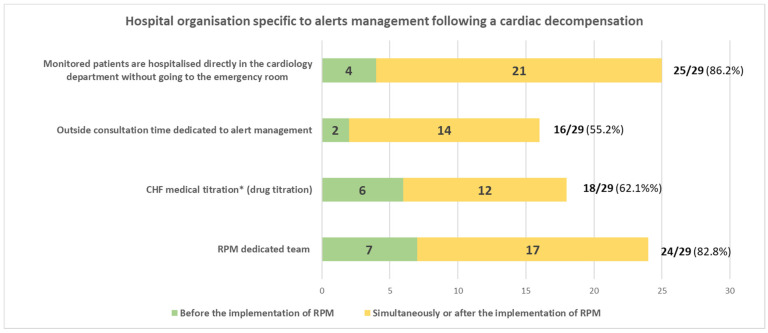
Specific organisational structures and procedures implemented in cardiology departments for alert management after decompensation of HF. * Titration of the dose of a drug in order to maximise its effectiveness without increasing the risk of adverse drug events, as recommended for patients with HF [11,12].

**Table 1 ijerph-20-04366-t001:** The criteria for organisational impacts in the health technology assessment defined by the HAS.

The HAS Criteria for Organisational Impacts		Potential Impact of CCC^TM^	
Macro-Criteria	Sub-Criteria	Evaluation Criteria	Data Source
**Macro-criterion 1:** Impacts of the health technology on the care PROCESS	1.1 Times prior to the initiation of the process	Time between cardiac decompensation and the medical care	S * O **
	1.2 Process pace or duration	Duration of the whole medical process T(0)-T(end): 1st consultation- 1st care/last consultation -last care	S O
		Number of cardiac decompensations	O
	1.3 Process timing or content	Modification of the timeline	S
		Modification of the content	S
		Changes and evolution of the content and the timeline	S
	1.4 Number or type of staff involved in the process: quantitative view of human resources	Presence of an allocated team for remote monitoring and allocated time	S
		Healthcare professionals involve in RPM outside of the hospital (ambulatory care)	S
	1.5 Type or frequency of use of products, devices, materials, equipment, infrastructures, and information systems used in the process	Allocated materials and equipment (at home, at hospital, for ambulatory care)	S
	1.6 Quality and safety of the environment or context in which the process takes place	New hospital organisation specific to alert management	S
		Impact on continuity of care	S
**Macro-criterion 2:** Impacts of the health technology on the abilities and skills required of stakeholders to implement the care process	2.1 Stakeholder’s required skills and expertise associated with the delivery or provision of care	Disease management for patient	S O
		Healthcare professionals training	S O
	2.2 Ability to share and transfer skills, knowledge, and know-how with other stakeholders	Professional delegation (from the cardiologists to the nurses) for the patient follow-up care	S
		Level of knowledge of the patient	S
		Ability to share information	S
	2.3 Scheduling and planning capacities for healthcare services or the patient or carer	New hospital organisation specific to alert management: Impact on care scheduling for the medical process of patients in critical condition Impact on the prioritisation of care regarding the medical risk Respect for the carers’classification	S
	2.4 Scheduling and planning capabilities between care structures or combinations of stakeholders	Inclusion of new non-hospital stakeholders (healthcare professionals from monitoring centre)	S
		Impact on the coordination between the ambulatory care and the hospital	S
	2.5 Stakeholders’ working or living conditions	Healthcare professionals’ perception on the evolution of their working conditions	S
		Patient’s quality of life	S
		Impact on the reduction of carers’ mental workload thanks to the alerts	S
	2.6 Terms, nature, or source of stakeholders’ funding	Impact on the budget	O
		Financing of RPM dedicated human resources at the hospital	O
		Financing of RPM dedicated material	O
**Macro-criterion 3:** Impacts of the health technology on society or the community	3.1 Impact on community in terms of health and safety	Not applicable ***	
	3.2 Impact on social inequalities or accessibility to care	Care access	O
		Impact on social and territorial inequalities	O
		Impact for low-income patients	O
	3.3 Impact on social or work relationships or in terms of society as a whole	Not applicable ***	
	3.4 Impact on environmental footprint	Not applicable ***	

* S: potential impacts of CCC^TM^ documented using the survey data. ** O: potential impacts of CCC^TM^ documented using other data sources (in-house data and other studies). *** Not applicable: potential impacts of CCC^TM^ that were not relevant to the survey or could not be transposed to the survey.

**Table 2 ijerph-20-04366-t002:** Baseline characteristics of the respondents.

	N = 29 Cardiology Departments	
Baseline Characteristics of the Respondents	n	%
Status		
Public-sector hospital	24	83%
Private-sector (not-for-profit) hospital	1	3%
Private-sector (for-profit) hospital	4	14%
Number of patients monitored *		
Mean (standard deviation)	122 (97.4)	
Median	80	
Time using Chronic Care Connect^TM^ (months)		
Mean (standard deviation)	23 (11.3)	
Median	24	
Less than 12 months	7	24%
12 to 24 months	8	28%
More than 24 months	14	48%

* between 1 February 2018 and 26 April 2021.

**Table 3 ijerph-20-04366-t003:** Alert management tasks performed by healthcare professionals.

	N = 29 Cardiologists Involved in Alert Management	Time Spent per Week (Mean)	N = 24 Nurses Involved in Alert Management	Time Spent per Week (Mean)
Acknowledgment of an alert	16 (55%)	2 h	23 (96%)	10 h
Diagnosis following an alert	23 (79%)		19 (79%)	
Response to an alert *	22 (76%)		22 (92%)	
Intervention following an alert **	26 (90%)		21 (88%)	

* a call from a patient, contacting the patient’s general practitioner, scheduling a consultation, scheduling hospital admission, asking for additional information, transferring to the monitoring centre, etc. ** changes in medication, disease management, etc.

## Data Availability

Data is unavailable due to privacy or ethical restrictions.

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
