# Peer review of "Organisational Impact of a Remote Patient Monitoring System for Heart Failure Management: The Experience of 29 Cardiology Departments in France"

_ijerph, 2023, doi:10.3390/ijerph20054366_

Round 1

Reviewer 1 Report

The authors should be congratulated for presenting a study of cardiology departments in France to describe the organizational impact of new Chronic Care Connect TM (CCCTM) system for remote monitoring of patients with HF.

However, I have several comments that limit the scientific value of the manuscript.

M comments:

1.       Please add percentages next to the numbers in the Abstract and in the Results, no need for decimals as the max number is only 29

2.       What are the conclusions drawn from the survey? Is there clinical impact on providing the service to the patients? Please add in the abstract

3.       Line 53, missing reference, pls add

4.       Lines 75-82, missing references, pls refer to the document accordingly

5.       Table 2, pls remove the time frame and reposition below the table

6.       The whole text needs minor polishing for English language (for example, “In the seven other…”, “six of the 18”, “25 of the 29”, “Two of the other five”, “mentioned five times”,

7.       Can the authors make Results section shorter? Pls put only relevant results in the text as you already have tables presenting the results and avoid repetition

8.       Pls start Discussion with main findings instead of findings from the other studies

9.       One of the main limitations is there is no scoring system in the questionnaire for assessment and the number of centers is relatively small. Pls add to the limitations

Author Response

Dear reviewer,

On the behalf of this article's authors, I would like to thank you for this review, which will certainly improve the quality of our article. 

Regarding your comments, we globally agree with you and tried to follow them. 

You will find below a detailed answer to each of them. 

Best regards,

Laurène COUROUVE. 

  1. The first recommendation on the percentages has been followed. 
  2. Regarding the following comment “What are the conclusions drawn from the survey? Is there clinical impact on providing the service to the patients? Please add in the abstract”. Unfortunately, it is not possible for us to address the clinical impact on providing the service to the patients. Our article focuses on assessing the organizational impact by using official guidelines which are not including the assessment of clinical impact. However, another study on the clinical impact of CCCTM is currently conducted. This study will complete the results presented in this article.
  3. The missing reference line 53 has been added. 
  4. The missing references lines 75-82 can’t be added. We didn't put any references, as these elements are from the study on which the article is based. Moreover, those elements refer to a report which is not published.  
  5. Regarding Table 2, we removed the time frame and repositioned it below the table, as it was asked. 
  6. The recommendations regarding the language have been followed, but we are not sure if the corrections match your expectations. 
  7. We tried to shorten the Results by following your recommendation.
  8. The Discussion now starts with the specificities of the study. 
  9. Indeed, the relatively small sample and the lack of scoring systems are both part of the limitations of our survey, thanks you for pointing them out. 

Reviewer 2 Report

Although the evaluation of the impact of a new system on an organization is very important, there are few papers that discuss it. In this sense, this report is very valuable and interesting.

However, there are few descriptions of CCC, and readers who are not familiar with this system will find it difficult to agree with the content of this paper.

I think the paper would be more interesting if there were  descriptions of what exactly the CCC system is and what it can do.

Author Response

Dear reviewer,

On the behalf of this article's authors, I would like to thank you for this review, which will certainly improve the quality of our article. 

Regarding your comment on the description of CCCTM, we agree with you, it will be more interesting and easier for readers if they have a better understanding of the device’s features. Thus, we added a more detailed description of CCCTM in the Method.

Best regards,

Laurène COUROUVE.